# Identification of the Mixing Ratios of Beer-Mixed Beverages by UV/VIS Spectroscopy

Sophia Stinus [1], Pauline Erban [1], Josef Diebold [1] and Magnus S. Schmidt [1,2,*]

1    Medical and Life Sciences Faculty, Furtwangen University, Jakob-Kienzle-Str. 17, D-78054 VS-Schwenningen, Germany; sophia.leonie.stinus@hs-furtwangen.de (S.S.); pauline.erban@hs-furtwangen.de.de (P.E.); josef.diebold@hs-furtwangen.de (J.D.)

2    Institute of Precision Medicine, Organic and Bioorganic Chemistry Labs, Furtwangen University, D-78120 Furtwangen, Germany

*    Correspondence: magnus.schmidt@hs-furtwangen.de

**Abstract:** UV/VIS spectroscopy was used in conjunction with a calibration method to determine the mixing ratio of samples of beer-mixed drinks with mineral water (sour Radler beers) from the food service industry. For this purpose, calibration lines were constructed based on absorptions at the two wavelengths 380 nm and 430 nm. To create these calibration lines, blending ratios were prepared from beers from three southwestern German brands and mineral water. The samples of the sour Radler beers were taken from restaurants and bars, which are all located in Baden-Württemberg. It has been shown that the calibration lines can be used to determine mixing ratios with an average deviation of approximately 7% and 9%. Thereby, the calibration line based on the absorption at 430 nm represents a higher accuracy. Using this calibration line, the sour Radler beer samples have percentage ranges of beer of approximately 50% to 90%. This method could serve as a rapid quantitative quality control of the mixing ratios of sour Radler or other beer-mixed drinks. It could be used in breweries for routine control.

**Keywords:** UV/VIS spectroscopy; photometry; beer; mixing ratio; color; food quality





## 1. Introduction

Beer is the most consumed alcoholic beverage in the world [1,2]. A German consumes on average 100 L per year [3]. Mixed beer drinks such as Radler beer are also becoming increasingly popular, thanks to their thirst-quenching effect and low alcohol content of 1 to 2.7% [4]. Radler consists of about 50% Pilsner beer and 50% lemonade (Radler sweet) or sparkling water (Radler sour). Pilsner beer is a bottom-fermented beer that is fermented and stored at low temperatures. The beer is filtered to remove yeast cells and turbidity contained in the beer in order to give it a clear appearance [4]. It is a strongly hopped beer, highly carbonated and frothy [5].

According to the German Purity Law, beer is made exclusively from barley malt, hops, yeast and water [6]. The characteristic coloration is due to the molecules extracted from the malt grain and hops. Malt production, which is the germination of the ripened grain, ends with kilning at 80 °C. At this high temperature, the Maillard reaction starts, which influences the coloring and aromas [7]. A chemical reaction takes place between the carbonyl groups of the reducing sugars and the amino groups of the amino acids, peptides or proteins [8]. Heterogeneous, nitrogen-containing, aroma intense brown pigments known as melanoids are formed, which absorb light at wavelengths as high as 420 nm [8]. Besides color, melanoids also contribute to the stabilization of aroma compounds and foam and register anti-oxidative properties [9]. In addition, reactions of amino compounds with dicarbonyls occur, which are called Strecker aldehydes [4]. These are usually very aroma intensive.

Also, color defining is the oxidation of polyphenols during storage and ageing [10]. Flavan-3-ol monomers and proanthocyanidin oligomers are the two main phenolic com-

ponents which influence the beer color changes as a result of oxidation [10]. Phenolic compounds are phytochemicals which are derived from phenylalanine and tyrosine [11]. They are secondary metabolites of plants [11]. Of the polyphenols in beer, 70–80% originate from the malt and 20–30% from the hops [12]. In addition to color, polyphenols also influence bitterness, flavor, odor, and oxidative stability of food items [11].

For the sensory quality, the color of beer is of great importance, as it gives the consumer the first impression. In many breweries, color determination is still done visually by color comparison with standardized color discs and indication of the color in units of the European Brewery Convention (*EBC*) scale [4]. However, this method is very subjective, as it depends on the examiner's vision and impression [10]. An alternative method that neglects the subjective impressions of humans is the use of ultraviolet and visible (UV/VIS) spectroscopy as a simple and reliable method to determine the color of beer. The UV-VIS range extends from 200–800 nm and the light absorption spectroscopy in the UV-VIS region is used to analyze a wide range of food and beverage samples [11]. The ultraviolet region ranges from 200 nm to 350 nm while wavelengths greater than 350 nm belong to the visual range [13]. This spectroscopic technique is frequently used in food analysis because it is a simple, accurate, fast, and high precision method [11]. According to the standard Analytica-*EBC* color measuring method (European Brewery Convention 1975) the absorbance to determine the color of beer is measured at 430 nm. A factor is used to adjust the absorption of the *EBC* color scale [3].

$$EBC = A_{430} \cdot 25$$

The *EBC* scale ranges from 0 to 80. As a rather light-colored beer, Pilsner beers typically have *EBC* values of 5.3 to 7.5 [14]. In the United States, the color of beer is also determined using a photometric measurement method at 430 nm called the Standard Reference Method (SRM). The *EBC* method is multiplied by a factor of 25, whereas the SRM is multiplied by a factor of 12.7. Thus, the color spectrum of the SRM takes values from 2 to 40 [15].

In the following, the absorption of sour Radler beer was determined photometrically to calculate their mixing ratio. For this purpose, calibration lines with known mixing ratios were created to be able to determine the dilution of unknown samples. This method could serve as a rapid quantitative quality control of the mixing ratios of sour Radler beer and could be used in breweries for routine control.

## 2. Material and Methods

### 2.1. Sample Collection of Radler Beer Samples from the Gastronomy

Bars, respectively restaurants, serving sour Radler beers with the indicated beer types (beer A, beer B, or beer C) were selected for sampling. The beers originate from the southwestern region of Germany and were manufactured traditionally according to the law of purity. A total of twelve samples were taken, four of which were mixed with beer A, beer B, or beer C. The sour Radler beer samples were numbered from 1 to 12 to keep them apart. The samples consist of an unknown mixing ratio, which was determined using the calibration lines established by photometric measurement at 380 nm and 430 nm.

### 2.2. Degassing

Due to the high carbonic acid content, the liquids, including the beers and the water, the standards of the specified mixing ratios and the samples of the gastronomy were degassed at the Schlenk line, which consists of a vacuum manifold and an inert gas manifold. A round-bottom flask with a contained magnetic stirring bar was filled with the liquid to be degassed. The liquid was cooled down in an ice bath. Since the liquid has a lower boiling temperature in a vacuum, evaporation of small amounts of the liquid was minimized. The construct of the round flask and ice bath was then positioned on a magnetic stirrer, which was set to 800 RPM to make sure that the liquid was evenly mixed. The round-bottom flask was connected to the Schlenk line with the aid of a transition piece and a rubber hose. The stopcock was closed during this process. To start degassing, the vacuum manifold was opened slowly and gradually. Foam formed, which rose in the round bottom flask.

When the foam had risen to the neck of the round flask, the valve was reset to the shut-off function. To remove the foam, the valve was set to the inert gas manifold for a few seconds. These two steps were repeated for about 15 min until the liquid was degassed.

The purpose of degassing is to prevent gas bubbles from forming in the cuvette that could cause a measurement error. In addition to vacuum degassing, it is also possible to degas liquids in an ultrasonic bath [16,17].

### 2.3. Sample Preparation of the Calibration Standards

To determine the mixing ratio of sour Radler beer in the gastronomic sector, calibration lines were created from Pilsner beers of the various southwest German brands and sparkling water (produced by a southwestern German brand, the "Classic" variety was selected, which is characterized by a high proportion of carbon dioxide). For this purpose, standards with a set of mixing ratios, which are defined in Table 1, were prepared. The Pilsner beers and the sparkling water were degassed before pipetting the defined mixture ratios. By photometrically measuring these standards, the linear relationship between the absorption and the percentage range of beer was determined.

**Table 1.** Defined mixing ratios of beer and water for creating the calibration lines.

| Percentage of Beer (Beer A, Beer B or Beer C) | 30 | 40 | 50 | 55 | 60 | 65 | 70 | 75 | 80 | 85 | 90 |
|---|---|---|---|---|---|---|---|---|---|---|---|
| Percentage of sparkling water | 70 | 60 | 50 | 45 | 40 | 35 | 30 | 25 | 20 | 15 | 10 |

### 2.4. Sample Preparation of Positive Controls

In addition, four defined mixing ratios of beer (beer A, beer B and beer C) and water were prepared with a ratio of beer to water of approx. 50:50, 60:40, 70:30, and 80:20. In contrast to the samples which were prepared for the calibration line, the positive controls were degassed only after the preparation of the mixture ratios. By measuring the samples with the known mixing ratios, the accuracy and the deviation of the various calibration lines were determined.

### 2.5. UV/VIS-Spectroscopy

For UV/VIS spectroscopy, the model "LAMBDA XLS+" from PerkinElmer (Waltham, MA, USA) was used in combination with the optical precision glass cuvettes from "HELLMA®" (Müllheim, Germany). The cuvettes correspond to type number 6030 and are characterized by a layer thickness of 10 mm. Three milliliters of each sample were added to each cuvette. Only degassed samples were placed in the UV/VIS spectroscope. The zero adjustment was always performed with the degassed sparkling water, and a triple determination was always conducted.

First, the spectra of beer A, beer B, and beer C were obtained in the range from 370 nm to 800 nm. The absorbance at 430 nm was measured to determine the *EBC* value of each beer. Subsequently, the standards with the mixture ratios from Table 1 were determined photometrically at a wavelength of 380 nm and 430 nm for each beer. Based on these absorptions at 380 nm and 430 nm, two calibration lines per beer were obtained. The prefabricated and degassed positive controls, as well as the twelve degassed samples from the gastronomy, were also measured photometrically at 380 nm and 430 nm. Microsoft Excel (Version 2211) was used to evaluate the experiment. The program applies a simple linear regression to create calibration curves.

## 3. Results and Discussion

### 3.1. Characterization of the Beers

In order to characterize the three different beers A, B, and C their *EBC* value was calculated from the absorption at 430 nm according to the standard Analytica-*EBC* color measuring method (European Brewery Convention 1975). Before measuring the absorption, the beer was completely degassed at the Schlenk line. Because Pilsner beers are clear

beers that are already filtered during the production process to remove turbidity, it is not necessary to filter them again before measuring the absorption. The measured absorption was multiplied by 25 to adjust it to the *EBC* color scale [3].

As shown in Table 2, the *EBC* values of the measured beers differ slightly. Beer C is the most intensely colored of the beers measured, while beer B is the lightest. The calculated *EBC* values are typical for Pilsner beers, which have as rather light-colored beers an *EBC* range of 5.3 to 7.5 [14].

**Table 2.** EBC value of the three pilsner beers A, B, and C from the southwestern region of Germany.

| Measured Beer | *EBC* |
|---|---|
| A | 6.55 |
| B | 5.28 |
| C | 7.43 |

UV/VIS spectra of beer A, beer B, and beer C were obtained in the range from 370 nm to 800 nm. Figure 1 shows that the spectra of beers A, B and C differ only slightly. They do not show a characteristic peak in the UV-Vis region. Instead, the absorption decreases exponentially with increasing wavelength. Based on the spectra, an additional second wavelength was selected because the absorption at 430 nm for the Pilsner beers used was already less than 0.5. Due to the dilution of the beers during the production of the sour Radler, the absorption is even lower. Based on this, 380 nm was chosen because the absorbance at this wavelength was almost 1 for all three beers and, therefore, calibration lines with a larger slope could be created at this wavelength.

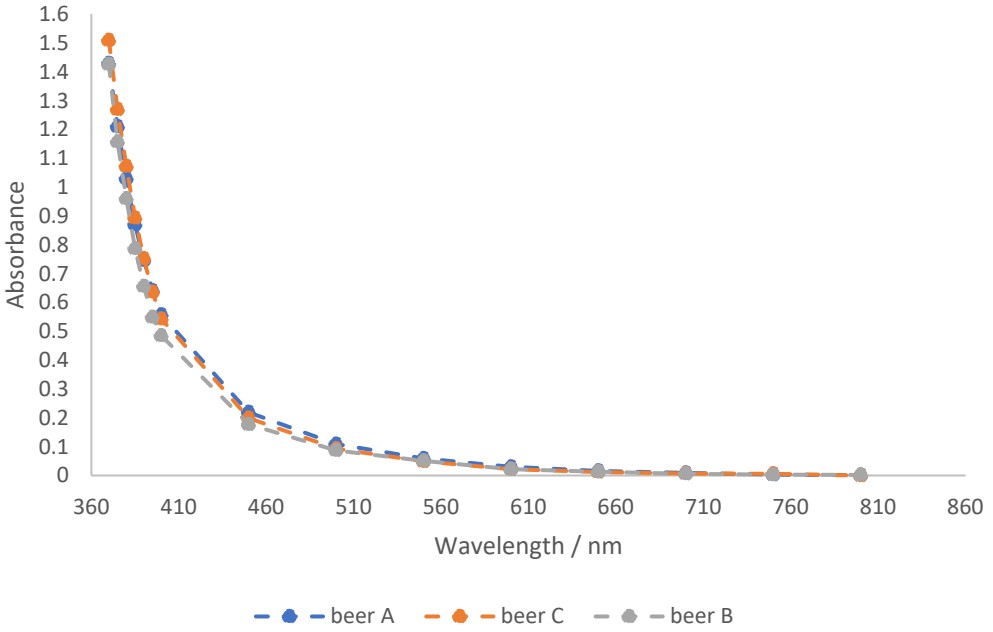

**Figure 1.** UV-VIS absorbance spectra of the three pilsner beers A, B, and C from southwest Germany.

### 3.2. Calibration Lines

With the help of the calibration lines, the determination of twelve Radler beer samples taken from the gastronomic sector was carried out. For this purpose, two calibration lines were created per beer. They were obtained with eleven different points in the percentage range of beer from 30 to 90. One is based on the absorption values measured at 380 nm and the other on those at 430 nm. These calibration lines of beer A, beer B, and beer C are shown in Figures 2–4. The calibration lines at 380 nm are steeper than those at 430 nm. This is also evident from the slope of the straight line equation of the various calibration lines.

The calibration lines whose absorption values were measured at 380 nm have a larger slope than the calibration lines at 430 nm. The slope characterizes the sensitivity. The steeper the slope of the calibration line, the more sensitive the analysis method [18]. Accordingly, the method of measuring the calibration line at an absorption of 380 nm is more sensitive than the method based on the absorptions at 430 nm.

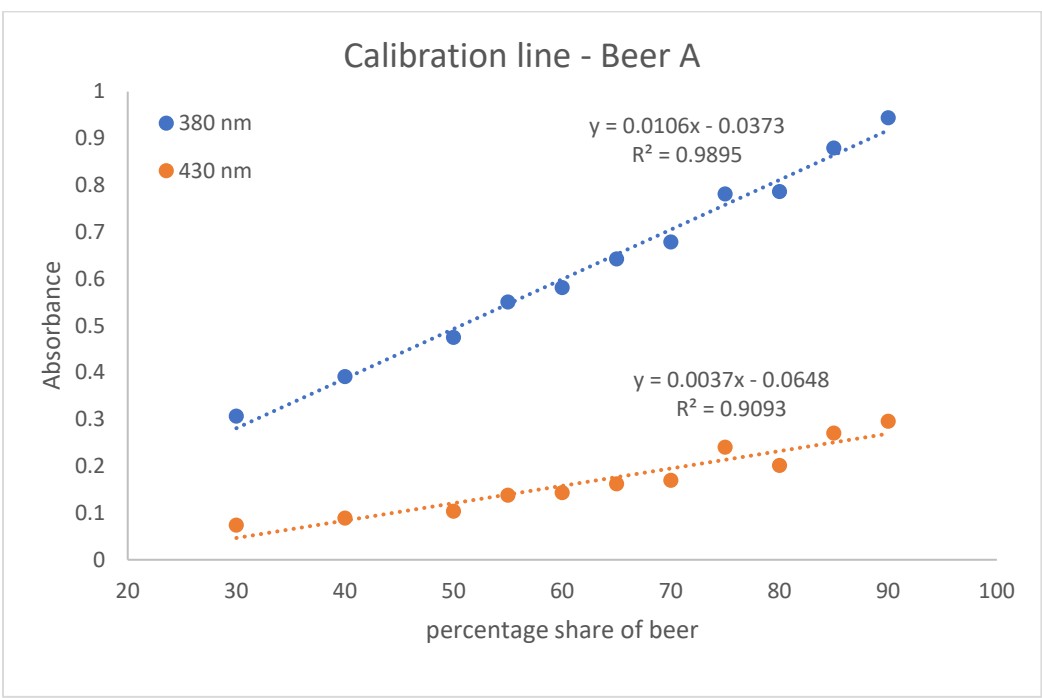

**Figure 2.** The calibration lines of beer A. The absorption is shown as a dependence on the percentage of beer. One calibration line is based on absorbance values measured at 380 nm and the other on absorbance values measured at 430 nm. The standard deviation for the measuring points of the calibration line at 380 nm ranges from 0.0012 to 0.0192. The standard deviation for the measuring points of the calibration line at 430 nm ranges from 0.0012 to 0.0156.

In addition, the coefficient of determination ($R^2$) for the calibration lines was determined. $R^2$ quantifies the correlation of the dependent variable with the independent variable by means of variance [19]. The $R^2$ of the calibration lines at 380 nm are closer to one. Thus, there is a greater correlation between the dependent variable (Absorption) and the independent variable (percentage range of beer) for the calibration lines at 380 nm. Nevertheless, it can be argued that both the calibration lines at 380 nm and the calibration lines at 430 show satisfactory measures of determination. On average, a coefficient of determination of 0.9935 is achieved for the calibration lines at 380 nm and a $R^2$ of 0.9397 for the calibration lines at 430 nm.

Due to this, the calibration lines, which are based on an absorbance at 380 nm, are probably more meaningful. In order to check the accuracy or deviation of the calibration lines and to test this hypothesis, four positive controls were prepared.

### 3.3. Accuracy of the Calibration Lines

Positive controls with known mixing ratios were prepared in-house to verify the accuracy of the calibration lines. For each beer A, B, and C, four dilutions were made with the mixing ratios (beer to water) of 50:50, 60:40, 70:30, and 80:20. This way, the calibration lines were to be checked in a wide range. Triplicate determinations were made, and the mean was substituted into the respective function of the calibration line, see Table 3, to calculate the associated mixing ratio. The measured absorbance was used for Y and the function was converted to X.

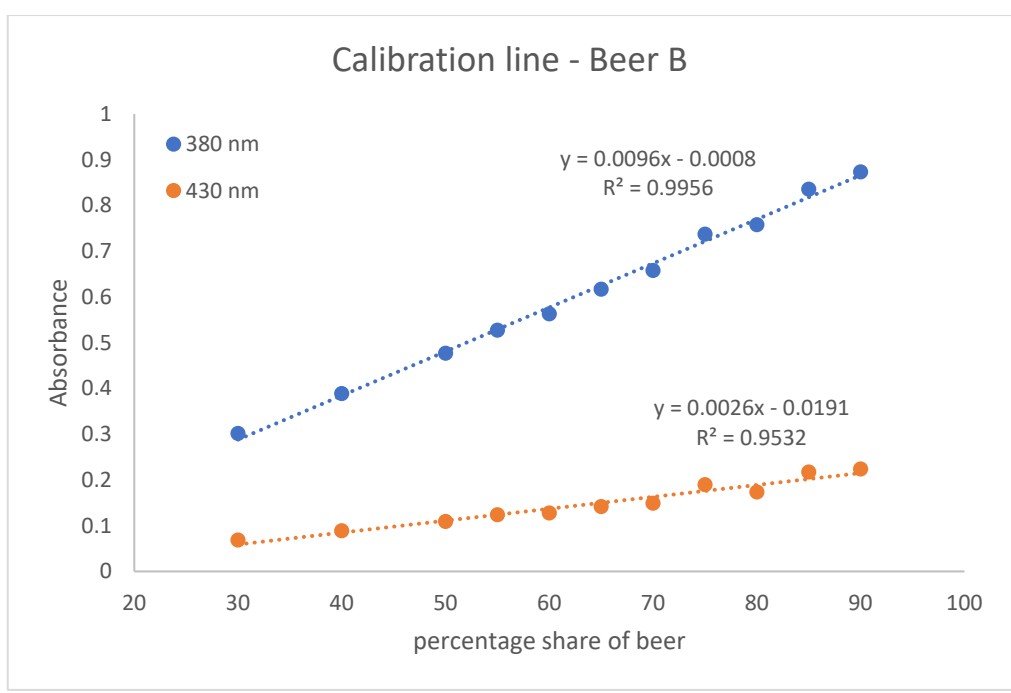

**Figure 3.** The calibration lines of beer B. The absorption is shown as a dependence on the percentage of beer. One calibration line is based on absorbance values measured at 380 nm and the other on absorbance values measured at 430 nm. The standard deviation for the measuring points of the calibration line at 380 nm ranges from 0.0017 to 0.0058. The standard deviation for the measuring points of the calibration line at 430 nm ranges from 0.0005 to 0.0038.

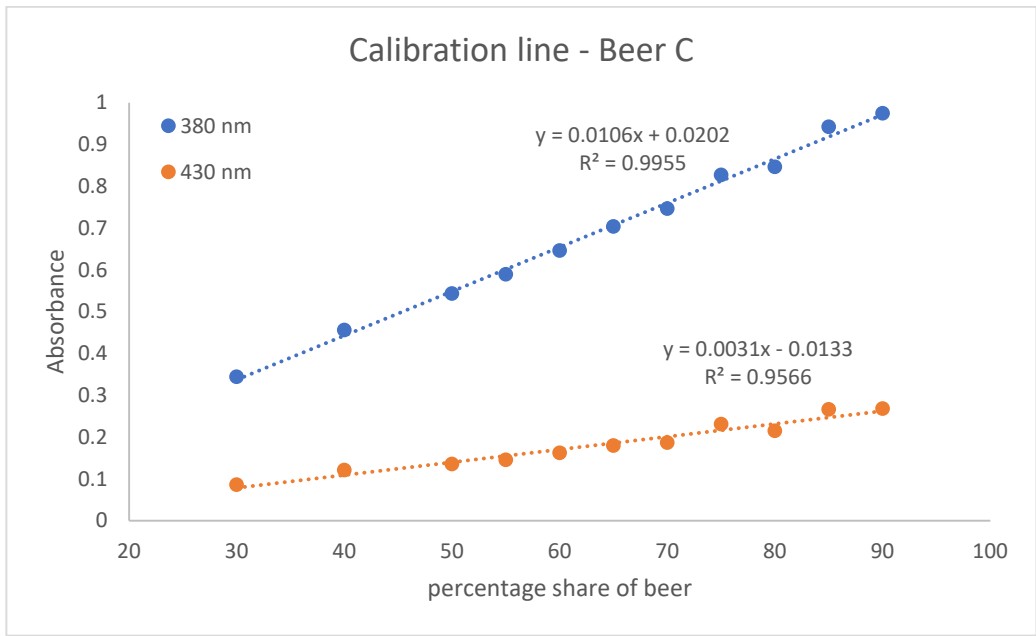

**Figure 4.** The calibration lines of beer C. The absorption is shown as a dependence on the percentage of beer. One calibration line is based on absorbance values measured at 380 nm and the other on absorbance values measured at 430 nm. The standard deviation for the measuring points of the calibration line at 380 nm ranges from 0.0012 to 0.0065. The standard deviation for the measuring points of the calibration line at 430 nm ranges from 0.0005 to 0.0042.

**Table 3.** Functions for calculating the Radler mixing ratios from the measured absorptions at 380 and 430 nm.

| Measured Beer | Function at 380 nm | Function at 430 nm |
|:---:|:---:|:---:|
| A | $x = \frac{y+0.0373}{0.0106}$ | $x = \frac{y+0.0648}{0.0037}$ |
| B | $x = \frac{y+0.0008}{0.0096}$ | $x = \frac{y+0.0191}{0.0026}$ |
| C | $x = \frac{y-0.0202}{0.0106}$ | $x = \frac{y+0.0133}{0.0031}$ |

The results, which are presented in Table 4, provided a very satisfactory correlation between the percentage of beer used to prepare the positive controls and the percentage calculated after measuring the absorbance using the functions of the calibration curves. The deviation between the calculated percentage of beer and the actual percentage used is on average 9.22% for the measurement at 380 nm, and ranges between 2.7% to 16.11% deviation. For 430 nm, the deviation between the calculated percentage of beer and the actual percentage used is 7.46% on average and ranges from 0.37% to 14.76% deviation. The values from the calibration line function at 430 nm, thus providing an even more accurate result.

**Table 4.** Percentage of beer of the self-prepared samples with known mixing ratios measured by 380 nm and 430 nm.

| Measured Beer | Percentage of Beer Used/% | Percentage of Beer Measured at 380 nm/% | Percentage of Beer Measured at 430 nm/% |
|:---:|:---:|:---:|:---:|
| A | 50 | 46.2 $\sigma = 0.1938$ | 48.7 $\sigma = 0.5554$ |
| | 60 | 54.4 $\sigma = 0.2705$ | 54.1 $\sigma = 0.3371$ |
| | 70 | 63.7 $\sigma = 0.3800$ | 63.3 $\sigma = 0.9187$ |
| | 80 | 73.0 $\sigma = 0.4513$ | 69.1 $\sigma = 0.8918$ |
| B | 50 | 55.1 $\sigma = 4.7209$ | 52.1 $\sigma = 0.7903$ |
| | 60 | 62.4 $\sigma = 0.3928$ | 62.4 $\sigma = 0.6281$ |
| | 70 | 71.9 $\sigma = 0.1964$ | 68.2 $\sigma = 0.3626$ |
| | 80 | 84.0 $\sigma = 0.4684$ | 80.3 $\sigma = 0.7903$ |
| C | 50 | 44.1 $\sigma = 0.2777$ | 47.0 $\sigma = 0.4023$ |
| | 60 | 52.2 $\sigma = 0.3800$ | 54.8 $\sigma = 0.9250$ |
| | 70 | 58.7 $\sigma = 0.1177$ | 59.7 $\sigma = 0.4023$ |
| | 80 | 69.3 $\sigma = 0.0455$ | 69.3 $\sigma = 0.4023$ |

The results prove that the standard Analytica-*EBC* color measuring method at 430 nm is also suitable for the determination of sour Radler beer. Although the dilution of the beer results in lower absorptions at 430 nm than for pure Pilsner beer and the calibration line therefore has a lower slope, the mixing ratio can be reliably detected. This is con-

sistent with "Brautechnische Analysenmethoden: Methodensammlung d. Mitteleurop. Brautechn. Analysenkommission (MEBAK)" [20], who describe that the analysis of mixed beer beverages in terms of color is analogous to the determination of the color of beer.

### 3.4. Identification of the Mixing Ratio of the Gastronomy Samples

To determine the mixing ratio of sour Radler beers in the food service industry, various calibration curves were created. Three beers of different southwestern German brands were used, and the samples with defined mixing rations were measured at two different wavelengths (380 nm and 430 nm). This has resulted in six different calibration lines. The mixing ratios of the twelve different samples from the catering industry were determined with the aid of the corresponding linear equations listed in Table 3. The results are shown in Figures 5–7. The percentage of beer varies depending on which calibration line was used. With the aid of the calibration line based on the absorbance values at 430 nm, higher percentages of the beer were determined in 8 of the 12 results compared to the calibration curve at 380 nm. The greatest agreement of the mixing ratios is obtained by using the calibration lines of Beer A. Here, three (sample 1, sample 3, and sample 4) out of four samples show a deviation of approximately one percent.

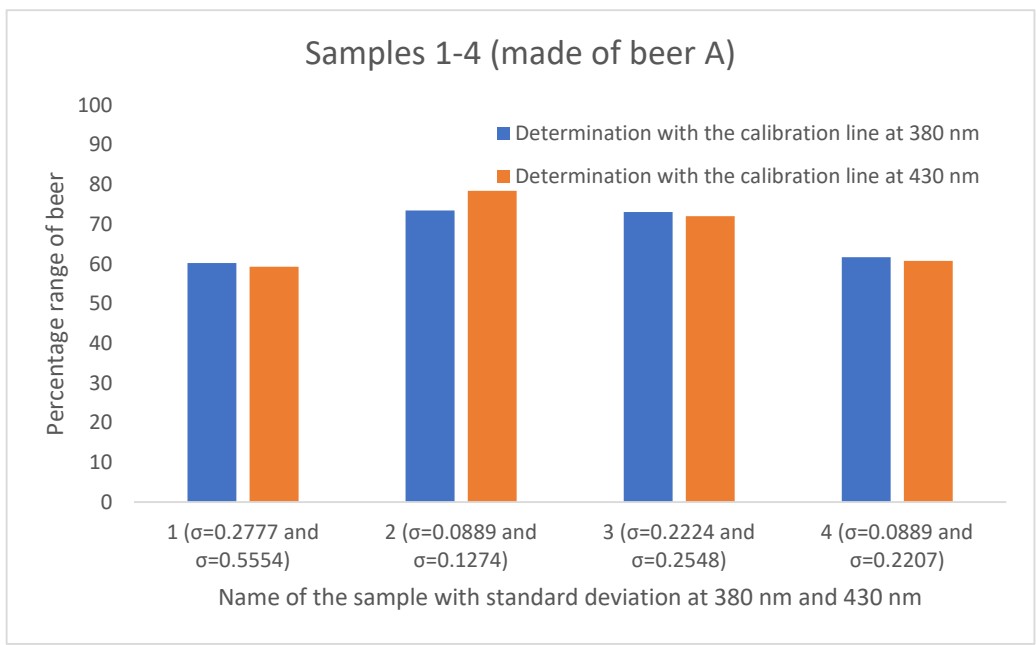

**Figure 5.** Comparison of the mixing ratios of the four samples of the gastronomy, prepared with beer A and established using the calibration lines based on absorbance at 380 nm (blue) and based on absorbance at 430 nm (orange). The standard deviation for the gastronomy samples ranges from 0.0889 to 0.5554.

The reason for the deviations of the mixing ratios is the use of different calibration lines measured based on absorptions of different wavelengths. Which result is more likely to be true cannot be verified, as the samples of the Radler beer were taken from the gastronomy. According to the European Brewing Convention, the color of beer is measured at a wavelength of 430 nm. Here, however, the spectra of the Pilsner beers show a low absorption, which results in an even lower value due to dilution with mineral water. Therefore, another wavelength of 380 nm was selected for photometric measurement since the beer spectra have higher absorption values at this point. The aim was to show whether a more sensitive calibration method (at 380 nm) gives more accurate results. However, this was refuted when the calibration line was checked using the positive controls, as the deviation based on the calibration line at 430 nm was 7.46% and based on the calibration line at 380 nm was 9.22% (Section 3.2). Thus, the straight-line equation of the calibration

line at 430 nm provides more accurate results, since for this the results have a smaller deviation on average.

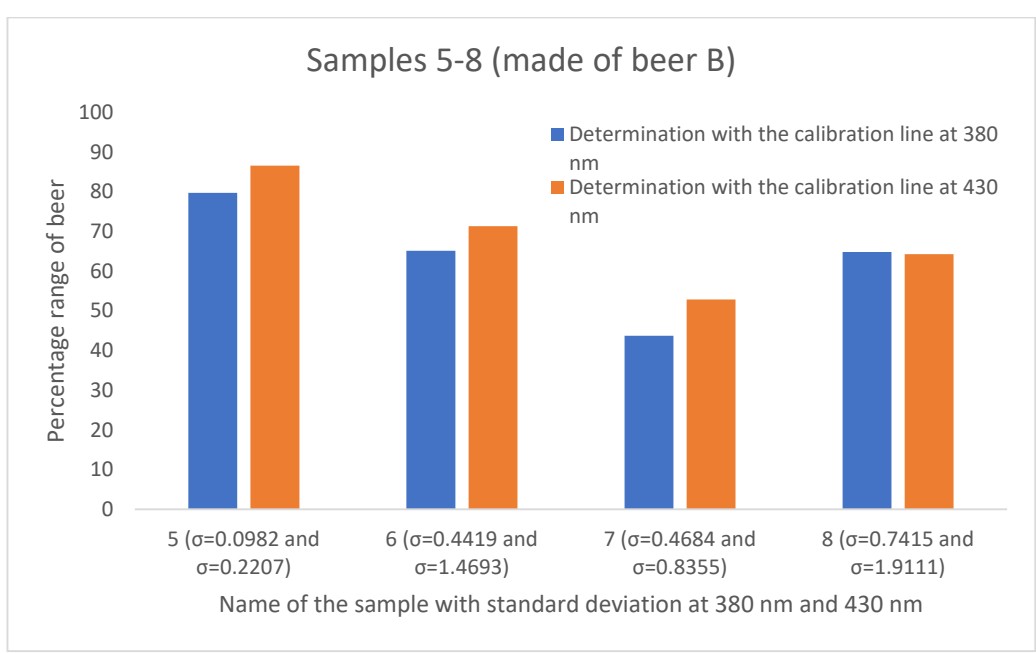

**Figure 6.** Comparison of the mixing ratios of the four samples of the gastronomy, prepared with beer B and established using the calibration lines based on absorbance at 380 nm (blue) and based on absorbance at 430 nm (orange). The standard deviation for the gastronomy samples ranges from 0.0982 to 1.9111.

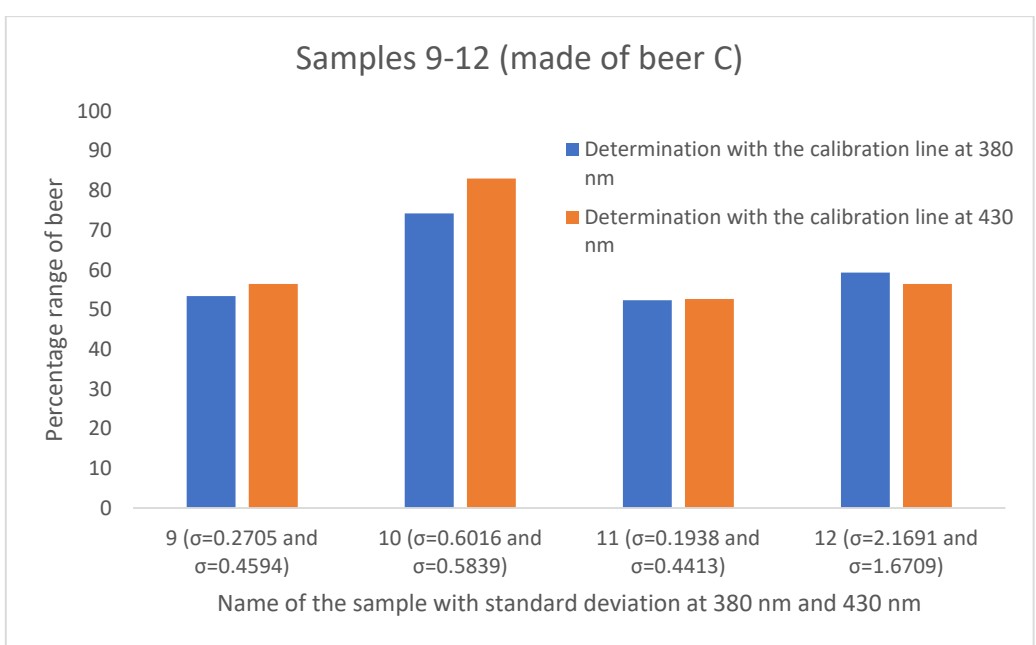

**Figure 7.** Comparison of the mixing ratios of the four samples of the gastronomy study, prepared with beer C and established using the calibration lines based on absorbance at 380 nm (blue) and based on absorbance at 430 nm (orange). The standard deviation for the gastronomy samples ranges from 0.1938 to 2.1691.

Sour Radler beer is a typical German beverage and it is a mixture between Pilsener beer and mineral water. The mixing ratio was approximately 50% Pilsner beer and 50%

mineral water. This mixing ratio should also be adhered to in gastronomy. Looking at the distribution of the results of the mixing ratios of the Radler beers determined with the straight-line equation of the calibration line at 430 nm (Figure 8), none of the 12 samples contained less than 50% beer. Most of the samples had a beer content of 50% to 60%. More than half of the samples, seven in total, were made with much more beer than specified in the definition. This speaks in favor of the restaurant industry, as they do not exhibit thriftiness in serving beer.

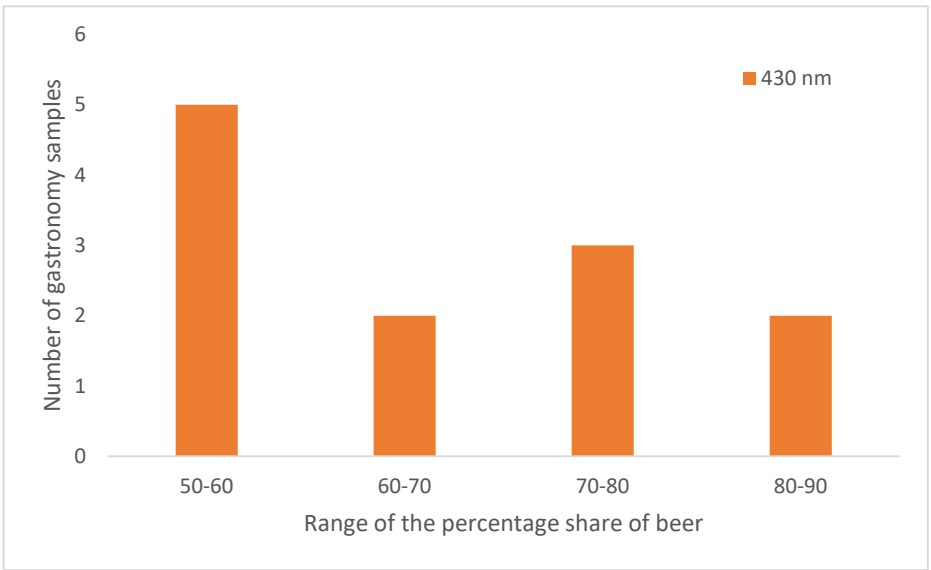

**Figure 8.** Distribution of the percentages of beer of the samples from the catering sector. The percentages of beer were determined using the straight-line equation of the calibration line based on the absorbance values at 430 nm.

A final aspect that could be useful in identifying, for example, is fake Radlers comprising yellow coloring to mask lower beer contents. This is shown in Figure 9. Here we calculated the absorbance ratio between the two wavelengths (A(380 nm)/A(430 nm)) and correlated it to the percentage share of beer. A higher deviation from these values could indicate the presence of corresponding colorings.

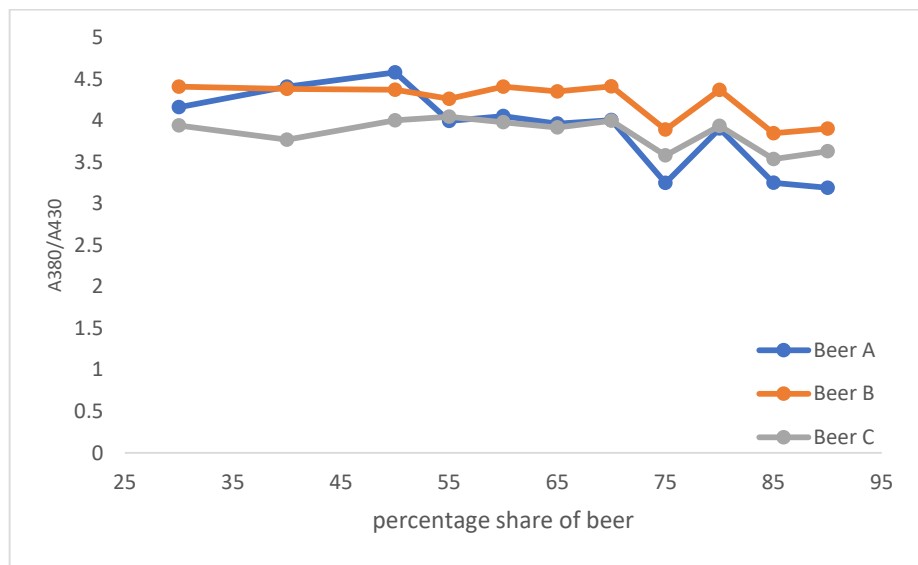

**Figure 9.** Absorbance ratio A(380 nm)/A(430 nm).

## 4. Conclusions

The calibration lines at 380 nm and 430 nm could determine the mixing ratio of the sour Radler beer samples with a deviation of approximately 7% and 9%. Therefore, the calibration line based on the absorption data of higher wavelength (430 nm) represents a higher accuracy. From these results, we can conclude that the *EBC*-method to measure the color of beer is also suitable for the beer-mixed drink, sour Radler. This method could serve as a rapid quantitative quality control of the mixing ratios of sour Radler beer and could be used in breweries for routine control. With adapted calibration lines, this method can also be used to determine the mixing ratios of other beer-mixed beverages as well as wine spritzers or juice spritzers.

**Author Contributions:** S.S. and P.E. performed the literature search. The manuscript was written by S.S. and P.E. The tables and images were produced by S.S. and P.E. Experiments were carried out by S.S. and P.E. The complete writing, design and corrections of the manuscript were led and supervised by J.D. and M.S.S. All authors have read and agreed to the published version of the manuscript.

**Funding:** This research received no external funding.

**Institutional Review Board Statement:** This article does not contain any studies with human participants or animals performed by any of the authors.

**Informed Consent Statement:** Informed consent not applicable.

**Data Availability Statement:** The datasets generated during and/or analyzed during the current study are available from the corresponding author on reasonable request.

**Acknowledgments:** We like to thank the ReAching program of the faculty MLS, Furtwangen University, for support.

**Conflicts of Interest:** All authors certify that they have no affiliations with or involvement in any organization or entity with any financial interest or non-financial interest in the subject matter or materials discussed in this manuscript. All Author declare that they have no conflict of interest.

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
