# Peer review of "Identification of the Mixing Ratios of Beer-Mixed Beverages by UV/VIS Spectroscopy"

_beverages, doi:10.3390/beverages9040092_

Round 1

Reviewer 1 Report

Comments and Suggestions for Authors

The manuscript “Identification of the mixing ratios of beer-mixed beverages by UV/VIS spectroscopy” is original and actual communication written in a clear form. The work considers spectral identification of beer-based mixtures. This study brings a contribution in field of food chemistry. The conclusions are useful for various food industry specialists.

 The study does not require any experimental revision, but the manuscript must be improved for better clarity:

1. Figure 1 contains a typo “Wavelenght”.

2. All Figures should be improved for better clearance: no frames and grid. Absorbance units to be declared.

3. The ratio of absorbance at two different wavelengths is also very useful, it can help to identify fake Radler (with lower beer contents) comprising yellow colorings. Thus, absorbance ratio at 380/430 nm must be given. If possible, the information about spectra of yellow food colorings can enrich the article — which of them can be detected using 380/430 nm ratio found in this study.

4. References must be numbered.

Author Response

The manuscript “Identification of the mixing ratios of beer-mixed beverages by UV/VIS spectroscopy” is original and actual communication written in a clear form. The work considers spectral identification of beer-based mixtures. This study brings a contribution in field of food chemistry. The conclusions are useful for various food industry specialists.

 The study does not require any experimental revision, but the manuscript must be improved for better clarity:

  1. Figure 1 contains a typo “Wavelenght”.

Corrected.

  1. All Figures should be improved for better clearance: no frames and grid. Absorbance units to be declared.

Frames and grids have been removed. We know that sometimes “AU” is used as absorbance unit. But since absorbance in general is unitless we decided to take that into account by not using AU as a “unit”.

  1. The ratio of absorbance at two different wavelengths is also very useful, it can help to identify fake Radler (with lower beer contents) comprising yellow colorings. Thus, absorbance ratio at 380/430 nm must be given. If possible, the information about spectra of yellow food colorings can enrich the article — which of them can be detected using 380/430 nm ratio found in this study.

Information has been added.

  1. References must be numbered.

corrected

Reviewer 2 Report

Comments and Suggestions for Authors

The authors established a new method for the identification of the mixing ratios of beer-mixed beverage. I thoroghly read the manuscript, and I think that the topic is interesting, well-organized, and well-written, which could attract the interest of beverages. However, some issues should be improved. I recommend major revsion.

Only the color comparison method is the current determination method for the sensory evaluation of beer? I mean, to beer-mixed beverages, is it necessary to determine the ratio between beer and lemonade or sparkling water? What is the reason for doing this? What is the method for the determination their ration currently?

Editing errors:

Line 34, please use Oxford comma, such as: a, b, and c, not a, b and c. Check the whole manuscript.

Line 51, the symbol “-”should be changd as “–”

Line 71, please number the equation.

Line 85–102, please combine this paragraph to 2.1–2.5.

Line 161, “…the three different …”, delete “the”, please check the whole manuscript.

Line 284, 9.22 %, there is no space between number and percentage unit (%), please check the whole manuscript.

The grammer, especially tense, should be improved.

Check the references’format.

Comments on the Quality of English Language

Moderate editing of English language required. The grammer, especially tense, should be improved.

Author Response

Reviewer 2:

The authors established a new method for the identification of the mixing ratios of beer-mixed beverage. I thoroghly read the manuscript, and I think that the topic is interesting, well-organized, and well-written, which could attract the interest of beverages. However, some issues should be improved. I recommend major revsion.

Only the color comparison method is the current determination method for the sensory evaluation of beer? I mean, to beer-mixed beverages, is it necessary to determine the ratio between beer and lemonade or sparkling water? What is the reason for doing this? What is the method for the determination their ration currently?

Mentioning sensory evaluation there are of course other methods including taste, aroma and so on. But our focus lies on an easy and fast instrumental method. Here also others could be mentioned, like nmr spectroscopy, but these are economically not interesting for industrial purposes and have therefore not been discussed in detail. Discussing the necessity to determin the ratio we discussed aspects like quality control, which is an important aspect of food production. Another aspect could be the evaluation of the quality of such beer-mixed beverages in gastronomy, were such mixed drinks are supposed to have at least an amount of 50% beer. In Germany for example such controls are performed by public control institutions, who work together with foodchemical labs which could perform the corresponding analytics.

Editing errors:

Line 34, please use Oxford comma, such as: a, b, and c, not a, b and c. Check the whole manuscript.

Corrected

Line 51, the symbol “-”should be changd as “–”

The technical editing of MDPI already worked the manuscript over and therefore we couldn’t see where a correction would be necessary.

Line 71, please number the equation.

Since this ist he only equation in the whole manuscript we did not see the necessity to number it.

Line 85–102, please combine this paragraph to 2.1–2.5.

Applied.

Line 161, “…the three different …”, delete “the”, please check the whole manuscript.

We are not quite sure about this comment, since the English editing partially added these “the” especially, so we kept it uncorrected.

Line 284, 9.22 %, there is no space between number and percentage unit (%), please check the whole manuscript.

Corrected to our best knowledge

The grammer, especially tense, should be improved.

English editing and check has been done and did not result in the necessity to improve the grammar

Check the references’format.

Done and corrected

Round 2

Reviewer 1 Report

Comments and Suggestions for Authors

All necessary edits were applied and the work can be published.

Reviewer 2 Report

Comments and Suggestions for Authors

Accepted as it is